# Enhancing Bread Quality with Steam-Treated Moringa (*Moringa oleifera*) Powder

**DOI:** 10.3390/foods14060927

**Published:** 2025-03-08

**Authors:** Takako Koriyama, Yuria Kurosu, Takahiro Hosoya

**Affiliations:** Faculty of Food and Nutritional Science, Toyo University, 48-1 Oka, Asaka-shi 351-8510, Saitama, Japan; s4c102400011@toyo.jp

**Keywords:** moringa leaf powder, steam treatment, loaf volume, antioxidant activity, quercetin-3-glucoside

## Abstract

Moringa leaf powder (MLP) is a nutrient-dense ingredient. However, its addition to bread often suppresses bread expansion, affecting its sensory properties. To address these challenges, this research explored how steam-treated MLP (SMLP) influences the expansion and sensory qualities of bread. MLP was steamed for 10 min in an electric oven, sieved, and incorporated at a 5% substitution level for wheat flour in bread formulations. SMLP improved the specific loaf volume, increasing it from 2.2 to 4.6 cm^3^/g compared to MLP. It also mitigated the inhibition of *Saccharomyces cerevisiae* (brewer’s yeast) viability induced by MLP, increasing its viability from 48% to 72%. Despite minor reductions, the antioxidant activity and quercetin-3-glucoside content remained high after treatment with SMLP. Moreover, SMLP delayed bread staling by reducing starch retrogradation enthalpy by 30–40%. Sensory evaluations revealed significant improvements in the aroma, appearance, and overall acceptability of bread prepared with SMLP compared to that prepared with MLP. This study demonstrated that steam treatment can enhance the potential and applicability of MLP as a functional food ingredient.

## 1. Introduction

Moringa oleifera, a species belonging to the Moringaceae family, is predominantly cultivated in tropical and subtropical regions [1]. In these climates, the leaves, seeds, and pods of moringa have been utilized as food sources for centuries. In particular, the tender foliage and early-stage pods of moringa are widely appreciated for their delicate texture and are commonly included in dishes such as green salads, curries, and stir-fries [2,3,4]. Moreover, the oil extracted from *M. oleifera* seeds is widely used for medicinal and cosmetic purposes, demonstrating its versatility [5,6]. Moringa leaves are recognized for their high nutritional content and provide a variety of essential vitamins, including vitamins A, several B-group vitamins (B1, B2, B6, and B12), vitamin C, and vitamin K, along with niacin, folic acid, and pantothenic acid. Additionally, they are rich in minerals, including potassium, magnesium, calcium, iron, zinc, and manganese. Therefore, they are a vital resource for addressing nutritional deficiencies, especially in developing countries. Recent studies have advanced our understanding of the functional properties of moringa leaves and seeds, reporting their antioxidant activity [7], anti-inflammatory effects [8], and blood sugar-lowering properties [9]. Specific bioactive compounds in moringa have also been reported to exhibit anti-inflammatory properties [10], promote insulin secretion [11], and possess antitumor and anticancer effects [12,13].

Moringa leaves, when processed into a powdered form, can be easily incorporated into food, offering a convenient means to enhance its overall nutritional value. Moringa leaf powder (MLP) has been successfully incorporated into various food applications, including bakery products, dairy alternatives, and functional beverages [14]. The addition of MLP enhances the nutritional profile of bread, cookies, and pasta by increasing protein, fiber, and antioxidant content, while also positively affecting sensory and textural properties [15]. These results highlight the viability of MLP as a naturally derived functional component in food development. Notably, the addition of MLP to wheat flour effectively fortifies bread with proteins, minerals, dietary fiber, and antioxidant components [16,17,18]. However, MLP significantly suppresses bread expansion, and its recommended usage is typically limited to 2–2.5% (*w*/*w*) to maintain acceptable bread quality [18,19]. Although adding high amounts of MLP could potentially lead to further improvements in the nutritional value of bread, practical challenges related to bread volume and texture have made it difficult to exceed a certain limit.

We have previously demonstrated that roasted MLP can inactivate the components responsible for suppressing bread expansion [20]. Specifically, bread with 5% untreated MLP had a loaf volume of 2.4 cm^3^ per gram of bread mass, whereas bread with 5% MLP subjected to dry heat treatment at moderate temperatures for a short duration exhibited an increased loaf volume, exceeding 4.0 cm^3^/g, indicating substantial improvement in expansion. Furthermore, bread with 5% MLP heated at a higher temperature for the same duration achieved a loaf volume comparable to that of MLP-free bread, reaching approximately 4.6 cm^3^/g. These results indicate that roasting enables the increased incorporation of increased MLP without compromising bread expansion. While concerns exist regarding the potential impact of MLP on gluten formation, our previous findings indicate that at 5% addition, MLP does not inhibit gluten network development. Instead, the reduced bread expansion is primarily attributed to the suppression of activity of *Saccharomyces cerevisiae* (brewer’s yeast) cells, which limits gas production [20]. Notably, this inhibitory effect was mitigated by heat treatment. However, the high dietary fiber and polyphenol content of MLP may affect other rheological properties, such as dough viscosity and water absorption. These factors should be considered when formulating MLP-enriched bread to optimize both texture and expansion.

While adding roasted MLP to bread is expected to enhance its nutritional value and functionality, challenges remain, such as the loss of the characteristic green color of the powder and changes in bread color. To address these issues, it is essential to identify methods that improve the expansion properties of bread while preserving its color and flavor. Roasting involves high-temperature dry heating and often leads to browning. In contrast, steaming, a moist heat treatment, could retain the vibrant green color of moringa powder while effectively acting on the components that suppress bread expansion. However, to date, no studies have investigated the effects of steam treatment on the expansion properties of MLP-containing breads. Moreover, while fortification using various bread improvers has been explored, no research has focused on improving expansion properties without the use of additional fortifiers. Verifying the effectiveness of steam treatment on MLP could provide a novel approach to enhance the quality of bread products.

Therefore, in the present study, we aimed to elucidate the effects of steam treatment on the expansion properties of bread containing moringa powder by considering factors such as the quercetin-3-glucoside (Q3G) content and the viability of *S. cerevisiae*. Additionally, we comprehensively evaluated the effect of steam treatment on MLP on the antioxidant activity, taste, texture, and anti-staling effects in bread. Notably, our study specifically investigates the ability of steam treatment to enhance bread expansion without the need for fortifiers, making it a novel approach to addressing the limitations of MLP supplementation.

## 2. Materials and Methods

### 2.1. Sample Preparation

MLP was obtained from Sun Rise, Ltd. (Okinawa, Japan). Steaming was performed for 10, 20, and 30 min in a steam mode in an electric oven (NE-BS604, Panasonic, Osaka, Japan). After steaming, the MLP was sieved through a no. 30-size mesh screen. For clarity, MLP that underwent steam treatment will hereafter be designated “steamed MLP” (SMLP). The prepared SMLP samples were placed in airtight containers and maintained at 4 °C under dark conditions until further analysis.

### 2.2. Bread Baking

The bread-making procedure followed a previously established method [20]. In formulations containing MLP, 5% of the total flour content was replaced with either MLP or SMLP. A control batch, prepared without MLP or SMLP, was also included for comparison. All necessary ingredients were sourced from a local supplier. After baking, the loaves were kept at 25 °C for 90 min to stabilize before conducting physical property measurements. Samples designated for further analysis were tightly wrapped in plastic film, enclosed in polyethylene bags, and stored at –30 °C until testing.

### 2.3. General Nutritional Composition Analysis

To evaluate the nutritional composition of wheat flour and MLP, key parameters such as moisture, crude protein, crude fat, ash, and crude fiber were analyzed in accordance with AOAC standard procedures [21]. Moisture contents were determined using an atmospheric drying technique at 135 °C, while crude fat content was extracted through a chloroform–methanol method. Crude protein analysis was conducted via the Kjeldahl method, employing a nitrogen-to-protein conversion factor of 6.25. Additionally, the Prosky technique was utilized to quantify dietary fiber content. Carbohydrate content was estimated based on the following Equation (1):(1)Total carbohydrate (g/100 g)=100−[moisture (%)+crude protein (%)+crude fat (%)+ash (%)

Energy values were calculated using the Atwater conversion system. For mineral composition analysis, samples were processed using the dry-ash preparation method. To extract mineral components, 2.0 g of the sample was combined with 50 mL of a 1.0% hydrochloric acid solution and stirred at 20 °C for 1 h. The resulting suspension was subjected to centrifugation at 3000 rpm for 15 min, and the supernatant was filtered through a 0.45 μm nylon membrane before measurement. The concentrations of sodium, potassium, calcium, magnesium, and iron were quantified using an atomic absorption spectrophotometer (AA-700; Shimadzu Co., Ltd., Kyoto, Japan). To prevent interference from calcium and magnesium ions, 0.1% strontium chloride was added to the samples prior to analysis.

### 2.4. Determining the Physical Properties of Bread

To assess the textural characteristics of bread samples, a TA XT Plus texture analyzer (Stable Micro Systems, Ltd., Surrey, UK) was employed. Bread slices were cut into uniform cubes measuring 2 cm × 2 cm × 2 cm. Each sample underwent a two-cycle compression test using a cylindrical probe (20 mm in diameter), operating at a speed of 1 mm/s until it reached a deformation of 70% strain. During the test, three key parameters indicative of crumb texture were recorded. Hardness was defined as the maximum force applied during the initial compression, while elasticity was measured as the ratio between the base lengths of the second and first compression curves. Chewiness was determined by multiplying hardness, cohesiveness, and springiness. For each loaf, six independent replicates were analyzed to ensure data reliability.

### 2.5. Color Measurements

The color intensity of the bread products was measured using a spectrophotometer (CM-700d, Konica Minolta, Tokyo, Japan), measuring the *L**, *a**, and *b** values according to the CIE color scale. *L** represents brightness from 0 (black) to 100 (white). The other two coordinates represent redness (+*a**) to greenness (−*a**) and yellowness (+*b**) to blueness (−*b**). The hue change (Δ*E*) caused by steaming was calculated using Equation (2):Δ*E* = [(*L*SMLP* − *L*MLP*)^2^ + (*a*SMLP* − *a*MLP*)^2^ + (*b*SMLP* − *b*MLP*)^2^]^1/2^(2)
All experiments were performed in triplicate.

### 2.6. Calculation of the Specific Volume of Bread

To evaluate the physical properties of the bread, its weight, loaf volume, and crumb texture were assessed. The loaf volume was measured using a 3D scanner (Ein Scan SP, Japan 3D Printer, Co., Ltd., Tokyo, Japan), where the turntable step was set to eight, and the mesh level was adjusted to high for precise measurement. The specific volume of the bread was then determined using the following Equation (3):*SV* = *V*/*m*(3)
where *SV* represents the specific volume (cm^3^/g), *V* is the loaf volume (cm^3^), and *m* is the bread mass (g). Each measurement was conducted at least in triplicate.

### 2.7. DPPH Radical Scavenging Activity

The antioxidant activity of the bread was evaluated using the DPPH radical assay. MLP (50.1 mg) and SMLP (50.0 mg) were mixed with 2 mL methanol to obtain the respective extracts. Moreover, bread samples were prepared, including control bread and bread supplemented with MLP or SMLP. After freeze-drying, the samples were mixed with methanol to obtain their respective extracts. The extracts were then centrifuged (13,000 rpm, 5 min, 20 °C), and the supernatant was collected. The solvent was evaporated using a rotary evaporator to obtain methanol extracts. Finally, the extracts were dissolved in DMSO at a concentration of 100 mg/mL and serially diluted with ethanol.

Subsequently, 100 µL of the diluted solution was transferred to each well of a 96-well microplate, and 100 µL of 0.4 mM DPPH radical solution was added. The reaction was carried out at 25 °C for 30 min. The absorbance was measured at 520 nm using a microplate reader (Synergy HTX, BioTek, Winooski, VT, USA). The residual DPPH radical percentage (%) was calculated relative to the control, and the 50% inhibitory concentration (IC50, mg/mL) was calculated.

### 2.8. Determination of Total Polyphenol Content

The total polyphenol content (TPC) was quantified using the Folin–Ciocalteu assay. A standard solution of gallic acid was initially prepared at a concentration of 100 mg/mL in DMSO and subsequently diluted with 80% methanol, yielding a series of dilutions ranging from 31.3 to 1000 µg/mL. For the reaction, 60 µL of water, 10 µL of the diluted gallic acid solutions, and 15 µL of Folin–Ciocalteu reagent were added to each well of a 96-well microplate, then left to react at 25 °C for 5 min. Following this step, 75 µL of a 2% sodium carbonate solution was introduced, and the mixture was incubated under dark conditions at 25 °C for 30 min. The absorbance of each sample was then recorded at 750 nm using a microplate reader (Synergy HTX, BioTek). A calibration curve was generated based on the absorbance readings and corresponding concentrations of the gallic acid standard. The TPC of the MLP and SMLP extracts was determined under identical conditions using 10 µL of each extract, prepared at a concentration of 3 mg/mL. The polyphenol content of the methanol extracts of MLP and SMLP was expressed as gallic acid equivalents (GAE, mg/mL), calculated from the absorbance values and the established calibration curve.

### 2.9. Quantitative Analysis of the Quercetin-3-Glucoside (Q3G) Content

High-performance liquid chromatography (HPLC) was performed using a pump (GL-7410, GL Sciences, Tokyo, Japan) and a UV detector (GL-7450, GL Sciences). The Q3G content was analyzed using Milli-Q water as solvent A and methanol as solvent B. An isocratic elution with 50% methanol (solvent B) was maintained for 10 min at a 1.0 mL/min flow rate, with an injection volume of 5 µL. Data acquisition was acquired by measuring absorption at 254 nm. A Q3G standard was serially diluted in methanol, and the area under the curve corresponding to a retention time of 5.45 min was used for quantification. A calibration curve was generated by plotting the peak area against the concentration of Q3G. Methanol extracts of MLP and SMLP were analyzed under identical HPLC conditions. Q3G quantification was performed by comparing the peak area to the calibration curve.

### 2.10. Determination of S. cerevisiae Viability

The viability of *S. cerevisiae* was measured using a Cell Counting Kit-8 (WST-8 assay) (Dojindo Laboratories, Kumamoto, Japan). A 10 mg/mL aqueous yeast (*S. cerevisiae)* solution was prepared by adding 10 mL of deionized water to 0.10 g of dry yeast (Nisshin Seifun Welna Inc., Tokyo, Japan). For each sample, 5 mg of MLP or SMLP was mixed with 1 mL of the aqueous yeast solution and incubated at 37 °C for 1 h. Subsequently, 350 µL of the supernatant was collected, and 7 µL of the WST-8 reagent was added, followed by incubation at 40 °C for 3 h. To account for background absorbance from the sample, 350 µL of the supernatant without WST-8 reagent was incubated under the same conditions. After incubation, all solutions were centrifuged (13,000 rpm, 5 min, 20 °C), and 100 µL of each solution was dispensed into each well of a 96-well microplate. The absorbance was measured at 450 nm using a microplate reader (Synergy HTX, BioTek). The viability (%) of *S. cerevisiae* was calculated by comparing the absorbance of the treated samples with that of the untreated control.

### 2.11. Retrogradation Enthalpy

The retrogradation enthalpy (Δ*H*) of the bread samples was determined using a differential scanning calorimeter (DSC 60 Plus, Shimadzu, Kyoto, Japan). To prepare the samples, the crumb portion of the bread was subjected to freeze-drying under vacuum conditions using a vacuum freeze-dryer (FDM-1000, EYELA, Tokyo, Japan). The process was conducted at a cold trap temperature of −54 °C and a vacuum pressure of 9.8 Pa for 24 h. After freeze-drying, approximately 10 mg of the dried bread sample was placed in an aluminum pan. Using a microsyringe, 20 μL of deionized water was added, ensuring complete hydration. The pan was then sealed and allowed to equilibrate overnight at 4 °C, enabling the water to be fully absorbed by the bread crumbs. For the DSC analysis, the sample pan was initially maintained at 20 °C for 30 min under isothermal conditions. It was then subjected to a heating program at a rate of 10 °C/min, with the temperature increasing from 20 °C to 100 °C. An empty aluminum pan was used as the reference. The onset temperature (*To*), peak temperature (*Tp*), conclusion temperature (*Tc*), and enthalpy change (Δ*H*) were derived from the endothermic peak of the thermal curve using Universal Analysis software (TA Instruments, New Castle, DE, USA). Each DSC measurement was conducted six times to ensure reproducibility.

### 2.12. Sensory Evaluation of Bread Samples

Sensory evaluation of bread samples was performed by twenty untrained panelists (five men and fifteen women, age range: 20–53 years). Prior to participation, all panelists provided informed consent, and the study was conducted in accordance with the principles outlined in the Declaration of Helsinki. Ethical approval for this research was granted by the Toyo University Ethics Committee (approval number: TU2022-010).

For the sensory assessment, each bread sample was cut into uniform slices measuring 2 cm × 2 cm × 2 cm. The samples were then placed in white plastic trays, each labeled with a randomly assigned three-digit code to ensure unbiased evaluation. To prevent flavor carryover between samples, participants were instructed to cleanse their palates with water before proceeding to the next sample. The panelists were asked to evaluate and assign scores to specific sensory characteristics, including taste, flavor, crust color, texture, and overall acceptability, using a nine-point hedonic scale. These sensory scores were then analyzed to determine the overall acceptability of the bread samples.

### 2.13. Statistical Analysis

Experimental data were processed using SPSS statistical software (ver. 27.0, IBM, Armonk, NY, USA). The data processing results were reported as the means ± standard deviations. DPPH radical scavenging activity, total phenolic content, quercetin-3-glucoside (Q3G) content, and sensory evaluation scores were analyzed using Student’s *t*-test to assess differences between samples. Student’s *t*-test was performed to compare specific sample pairs, with statistical significance set at *p* < 0.05 and *p* < 0.01. When multiple comparisons were required, the Bonferroni test was used for color and texture measurements, and Tukey’s test was used for bread staling, with statistical significance defined as *p* < 0.05.

## 3. Results and Discussion

### 3.1. Effect of Steam Treatment on Bread Expansion

Steaming MLP before incorporating it into bread significantly improved bread expansion compared to untreated MLP. As presented in Figure 1, bread prepared with untreated MLP exhibited a decreased expansion ratio. In contrast, bread prepared with MLP steamed for 10 min exhibited a markedly increased bread volume. However, extending the steaming duration to 20 or 30 min did not result in further improvements, suggesting a plateau effect.

Steaming alters the physical and chemical properties of MLP, which may contribute to improved dough expansion. As the steaming process was conducted under atmospheric pressure, the temperature did not exceed 100 °C. Therefore, the observed improvements in bread expansion were achieved without exposure to excessively high temperatures, which could otherwise cause thermal degradation of key MLP components. The lack of additional benefits with longer steaming durations suggests that a 10 min treatment is sufficient to optimize the effect of MLs on bread quality.

### 3.2. General Nutritional Composition of SMLP

The general nutritional composition of SMLP is presented in Table 1. The nutritional compositions of wheat flour and untreated MLP are also presented for comparison. Owing to differences in moisture content, the nutrient values were calculated on a dry-weight basis.

Both MLP (27.7 ± 8.2%) and SMLP (23.1 ± 0.6%) exhibited significantly higher protein content than that of wheat flour (13.5 ± 0.4%). Notably, the protein content of MLP was more than twice that of wheat flour, suggesting that the leaves are a rich source of protein. This high protein content highlights the potential of moringa as a nutritional supplement, particularly in regions suffering from malnutrition. Carbohydrate content was lower in MLP (53.2 ± 8.0%) and SMLP (60.2 ± 1.0%) than in wheat flour (84.6 ± 1.1%). This result confirms that moringa leaves are rich in proteins, lipids, and minerals, with a relatively low proportion of carbohydrates [3,4,5]. Crude fat content was significantly higher in MLP (8.8 ± 1.1%) and SMLP (7.8 ± 1.3%) than in wheat flour (2.6 ± 0.8%). This result confirms that moringa leaves are lipid-rich and have the potential as a source of essential fatty acids [22]. The ash content, an indicator of mineral levels, was also significantly higher in MLP (10.8 ± 0.1%) and SMLP (8.8 ± 1.4%) samples than in wheat flour (0.5 ± 0.1%). Among the individual minerals, potassium (K) and calcium (Ca) were particularly abundant in MLP and SMLP. K content was 2570 ± 26 mg/100 g of MLP and 2377 ± 15 mg/100 g of SMLP, while Ca content was 2024 ± 22 mg/100 g of MLP and 1780 ± 10 mg/100 g of SMLP. These results suggest that moringa is a promising food source for addressing mineral deficiencies.

Although steaming resulted in a slight decrease in the levels of some nutrients, such as proteins and minerals, the overall nutritional value of the MLP samples remained largely unaffected. Importantly, high levels of key minerals, such as K and Ca, were retained, indicating that steaming effectively preserved most of the nutritional properties of moringa.

### 3.3. Color Properties of MLP and Bread After Steam Treatment

The color properties of SMLP and the bread prepared with it are depicted in Table 2. Both MLP and SMLP had significantly lower *L** values than those of wheat flour (*p* < 0.05), with SMLP having the lowest value (*L**: wheat flour, 78.0 ± 0.01; MLP, 55.3 ± 0.01; SMLP, 41.5 ± 0.01). The *a** values for MLP and SMLP were both negative, indicating a strong green hue, with MLP exhibiting a more pronounced green color (*a**: wheat flour, –1.1 ± 0.04; MLP, –7.0 ± 0.13; SMLP, –3.6 ± 0.04). This trend in *a** values was attributed to the presence of chlorophyll in the moringa leaves [23,24]. After steaming, SMLP exhibited a slight decrease in lightness (*L**) and a marginal reduction in green intensity (*a**). However, it retained the characteristic vibrant green color of moringa. Both the crumb and crust colors of MLP- and SMLP-containing breads exhibited significantly lower *L** values than those of wheat flour bread (*p* < 0.05). In particular, MLP-containing bread exhibited a pronounced darkening effect. This suggests that adding moringa strongly influenced bread color. The *L** value of wheat flour bread crumbs (78.1 ± 0.96) was significantly higher than those of MLP bread (41.7 ± 2.01). However, SMLP bread (51.8 ± 1.18) maintained a relatively brighter color than MLP bread. Similarly, SMLP bread demonstrated higher crust *L** values than those of MLP bread, indicating that it retained a bright overall appearance while preserving the green characteristics of the moringa (Figure 2).

These findings suggest that the color changes in moringa-supplemented bread are strongly influenced by the properties of the added powder. Specifically, SMLP effectively suppressed darkening and maintained a bright color while retaining the characteristic green hue of moringa.

### 3.4. Expansion and Physical Properties of Moringa-Containing Bread

The expansion and physical properties of bread containing MLP and SMLP were compared with those of wheat flour bread, and the results are presented in Table 2, with the appearance presented in Figure 2. The loaf volume (1780 ± 83.7 cm^3^) and specific volume (4.6 ± 0.01 cm^3^/g) of wheat flour bread were significantly reduced when MLP was added, at 914 ± 6.25 cm^3^ and 2.2 ± 0.1 cm^3^/g, respectively. These values were approximately half of those of wheat flour bread. This poor expansion led to inadequate crust formation and a sunken top of the bread. Additionally, the hardness of MLP bread (4.6 ± 0.86 N/cm^2^) was approximately five times higher than that of wheat flour bread (0.9 ± 0.23 N/cm^2^). However, its elasticity (0.1 ± 0.01) and chewiness (0.6 ± 0.03) were significantly lower than those of wheat flour bread. These results indicate that bread prepared with 5% MLP exhibited significantly inferior expansion and physical properties, which aligns with our previous findings [20] and those by Bourekoua et al. [22]. In contrast, bread containing SMLP exhibited a comparable loaf volume (1837 ± 21.6 cm^3^) and specific volume (4.6 ± 0.1 cm^3^/g) to wheat flour bread, overcoming the poor expansion observed in MLP bread. Moreover, the hardness of SMLP bread (0.5 ± 0.15 N/cm^2^) was lower than that of wheat flour bread, while its elasticity and chewiness (1.6 ± 0.45) were the highest among all samples. These results clearly demonstrate that steaming improved the properties of MLP, significantly enhancing both the expansion and physical properties of bread.

Typically, the gluten present in wheat flour absorbs water and interacts to form a network structure within the dough, which is critical for expansion. Insufficient water availability for gluten formation can result in poor expansion. However, in the present study, the moisture content of all bread samples was consistent at approximately 45% (Table 2), and no differences in water evaporation during baking were observed. Therefore, the water absorption capacity of MLP is unlikely to be directly involved in suppressing dough expansion. However, the reduction in the height and volume of MLP-containing bread is likely attributed to the antimicrobial properties of *M. oleifera* leaves. This antimicrobial effect may inhibit *S. cerevisiae* viability and alcohol fermentation during proofing [23]. In our previous study, roasting MLP at 130 °C for more than 20 min effectively alleviated this *S. cerevisiae* fermentation inhibition [20]. Similarly, the steaming process used in the present study may have inactivated the compounds that inhibit *S. cerevisiae* fermentation.

These results demonstrated that steaming improved the properties of MLP, significantly enhancing bread expansion and physical properties. While this study examined a simple, small-scale steam treatment method, its industrial scalability remains to be explored. Unlike roasting, steaming does not require high-temperature dry heating, making it accessible for small-scale applications. Future research should investigate its adaptability for commercial bread production.

### 3.5. Antioxidant Activity and Total Phenol Content (TPC) of Steamed MLP

The antioxidant activity of SMLP was evaluated using the DPPH radical scavenging method. As presented in Table 3, the IC_50_ values of MLP and SMLP were 318.9 ± 7.1 μg/mL and 439.9 ± 6.7 μg/mL, respectively, suggesting a slight reduction in antioxidant activity after steam treatment (*p* < 0.01). However, SMLP retained a relatively high antioxidant capacity, suggesting that steam treatment did not significantly compromise the antioxidant properties of the MLP.

The TPC also decreased significantly from 7.0 ± 0.2 μg GAE/mL in MLP to 6.0 ± 0.2 μg GAE/mL in SMLP (*p* < 0.01). This decrease may be attributed to the partial decomposition or denaturation of phenolic compounds in the moringa leaves caused by steam treatment. Nevertheless, the relatively high antioxidant activity of SMLP, despite its reduced TPC, suggests that non-phenolic components may also contribute to its antioxidant properties.

### 3.6. Q3G Content and Its Role in Antioxidant Activity

The content of Q3G, a major antioxidant compound, was analyzed in MLP and SMLP. The Q3G content decreased from 89.6 ± 0.65 µg/100 mg in MLP to 60.5 ± 3.85 µg/100 mg in SMLP. However, this reduction was not as pronounced as the loss observed in our previous study involving roasting at MLP 170 °C for 20 min [20], where the Q3G content decreased to less than one-fifth of the original level. The relatively high retention of Q3G in the SMLP suggests that steam treatment has a milder effect on Q3G than roasting.

Q3G is known not only for its antioxidant activity but also for its diverse functional properties, such as antitumor, anti-obesity, and anti-inflammatory effects [22]. Maintaining its content is essential for leveraging moringa as a food source. In this study, the antioxidant activity of Q3G was retained to a certain extent after steam treatment. This indicates that Q3G likely plays a significant role in maintaining the antioxidant properties of SMLP.

The processing of moringa powder through steam treatment minimizes the loss of antioxidant compounds, including Q3G, while maintaining key characteristics such as antioxidant activity and color. These results suggest that steam treatment is a promising technique for improving the quality of moringa as a food source, making it suitable for broader applications in functional food development.

### 3.7. Determination of S. cerevisiae Viability

The effects of steam treatment of moringa powder on *S. cerevisiae* viability were also evaluated, and the results are presented in Figure 3. When the *S. cerevisiae* viability of the control group (no moringa added) was set to 100%, the viability of the group with MLP added decreased significantly to approximately 48%. In contrast, the group treated with SMLP exhibited a recovery in *S. cerevisiae* viability to approximately 72% (*p* < 0.05). These results indicate that steam treatment of MLP mitigated the inhibitory effects of certain compounds on *S. cerevisiae* viability.

The substantial reduction in *S. cerevisiae* viability in the MLP group was likely due to the specific chemical compounds present in *M. oleifera* leaves, such as polyphenols and antimicrobial alkaloids [23]. These compounds have been suggested to affect the *S. cerevisiae* cell membranes and interfere with fermentation. In contrast, the recovery of *S. cerevisiae* viability to 72% in the SMLP group suggests that steam treatment caused the thermal decomposition or chemical modification of these inhibitory compounds, reducing their toxicity to *S. cerevisiae*. Furthermore, in the MLP group, the decrease in *S. cerevisiae* viability resulted in insufficient alcohol fermentation, significantly suppressing bread expansion (Table 2). In contrast, in the SMLP group, the improvement in yeast viability enabled normal fermentation, resulting in loaf and specific volume values comparable to those of wheat flour bread. These findings suggest that steam treatment is an effective processing method for partially inactivating *S. cerevisiae* inhibitory compounds while retaining the functional components of moringa.

However, it should be noted that steam treatment did not completely eliminate the inhibitory effects on *S. cerevisiae* viability, as approximately 28% of the *S. cerevisiae* cells failed to maintain viability. The remaining inhibitory effects may be attributed to polyphenols or alkaloids in the moringa leaves, which persist even after heat treatment. Further studies are required to identify these inhibitory compounds and optimize the processing conditions.

### 3.8. Bread Staling Evaluation

Bread staling was evaluated using DSC, and the temporal changes in enthalpy (ΔH) over a storage period of 0 to 7 days are presented in Figure 4. On day 0 (immediately after baking), no endothermic peak was observed at 50–65 °C in any of the samples. This indicates that the starch in the bread dough was fully gelatinized during baking and did not undergo recrystallization within 2 h after baking. These results are consistent with the findings of Salinas et al., who suggested that starch in freshly baked bread remains in a stable amorphous state [25]. However, starting from day 1 of storage, an endothermic peak was observed at approximately 50–65 °C in all samples, and ΔH consistently increased with the duration of storage. This result indicates that starch recrystallization or retrogradation occurred during storage, reflecting the progression of staling. On day 7, the control bread (without moringa added) exhibited the highest ΔH (an increase of approximately 3.4 J/g from day 0), confirming significant starch retrogradation. In contrast, bread containing MLP or SMLP demonstrated suppressed ΔH increases throughout the storage period. On day 7, the ΔH values of MLP and SMLP bread were 2.2 J/g and 2.0 J/g, respectively, representing a 30–40% reduction compared to that of the control bread (*p* < 0.05). Although no significant difference was observed between MLP and SMLP, the SMLP bread exhibited slightly lower ΔH values than those of the MLP bread. These results demonstrate that moringa powder, regardless of steam treatment, can inhibit starch recrystallization in bread. This inhibition may be attributed to the phenolic compounds in moringa leaves. Additionally, adding moringa powder may have improved water retention in the dough, delaying starch recrystallization. However, further studies are required to elucidate the detailed mechanisms underlying these effects.

The results of the present study indicate that bread with moringa powder, regardless of steam treatment, can significantly suppress starch staling compared to bread prepared without moringa powder. This suggests that moringa powder is a valuable food source for extending the shelf-life of bread and enhancing its storage stability.

### 3.9. Sensory Evaluation

The results of the sensory evaluation of bread containing MLP and SMLP are presented in Table 4. The evaluation criteria included taste, aroma, appearance, texture, and overall acceptability, which were assessed on a nine-point hedonic scale.

In terms of taste, SMLP bread (7.0 ± 1.4) received a higher score than MLP bread (5.7 ± 1.9), although the difference was not statistically significant. This may reflect the flavor-enhancing effect of steam treatment, although variability in individual sensory perceptions might have influenced the results. Regarding aroma, SMLP bread (6.5 ± 1.4) was rated significantly higher than the MLP bread (4.9 ± 1.6) (*p* < 0.01). This suggests that steam treatment reduced the distinctive unpleasant odor of moringa, resulting in a more favorable aroma. Similarly, the SMLP bread (7.5 ± 1.1) scored significantly higher than the MLP bread (6.2 ± 1.5) in terms of appearance (*p* < 0.05). This result indicates that steam treatment improved the visual appeal of the bread and enhanced its visual desirability. Although no significant difference was observed in texture, the SMLP bread (7.3 ± 1.6) received slightly higher scores than those of MLP bread (6.6 ± 1.7). The “chewy” texture of MLP bread and the “fluffy” texture of SMLP bread were distinct characteristics, both contributing positively to consumer preferences. In terms of overall acceptability, the SMLP bread (7.3 ± 1.3) was rated significantly higher than the MLP bread (6.0 ± 1.6) (*p* < 0.05). These results demonstrate that adding steam-treated moringa powder not only improved the expansion properties of the bread but also enhanced its sensory attributes.

These findings suggest that steam-treated MLP improves the appearance, aroma, texture, and overall acceptability of bread, making it a valuable ingredient for developing consumer-preferred products. Future studies should explore the application of moringa in other food products to develop new functional foods that meet consumer demands.

## 4. Conclusions

In the present study, we evaluated the effects of steam treatment (wet heating) on MLP on the expansion properties, quality, and functionality of bread. The addition of SMLP, which was subjected to 10 min of steaming, significantly improved the specific volume of the bread from 2.2 to 4.6 cm^3^/g, resulting in a softer and more elastic texture. We also revealed that steam treatment substantially alleviated the inhibitory effects of MLP on *S. cerevisiae* viability, thereby improving the expansion properties of bread. Furthermore, the antioxidant activity and Q3G content of SMLP were maintained at high levels, and an anti-staling effect on starch retrogradation was confirmed. The sensory evaluation demonstrated that bread containing SMLP exhibited superior appearance, flavor, and texture compared to bread prepared with untreated MLP. This highlights the enhanced quality and overall acceptability of SMLP-containing bread.

Collectively, our findings suggest that a brief 10 min steam treatment can significantly enhance the versatility of moringa, expanding its potential applications in various food products. Beyond bread, SMLP could be incorporated into other bakery products, such as cakes, cookies, and pastries, where its functional and nutritional properties may improve texture and sensory attributes. Given its ability to promote dough expansion while maintaining antioxidant properties, SMLP presents opportunities for developing novel functional bakery items. Further research is required to explore its large-scale processing potential and industrial adaptability to optimize its application in commercial food production.

## Figures and Tables

**Figure 1 foods-14-00927-f001:**
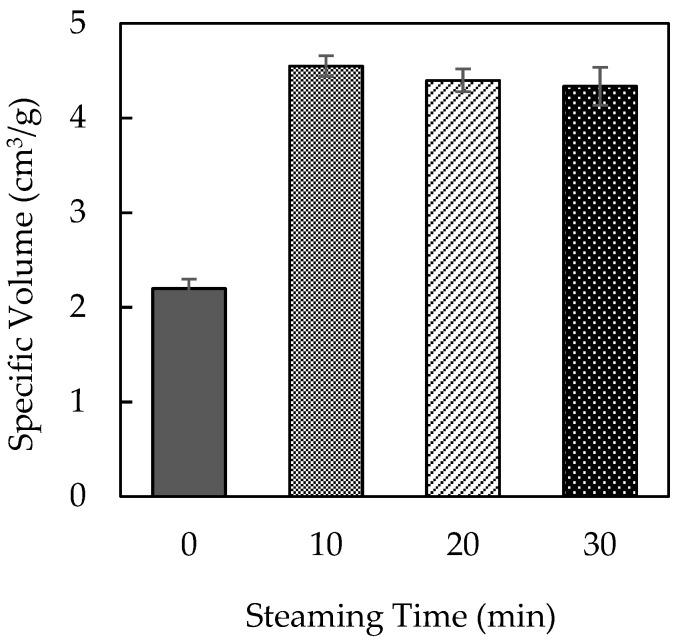
Bread expansion with steam-treated moringa leaf powder. The specific volume of bread baked with 5% moringa leaf powder after steaming for a designated duration.

**Figure 2 foods-14-00927-f002:**
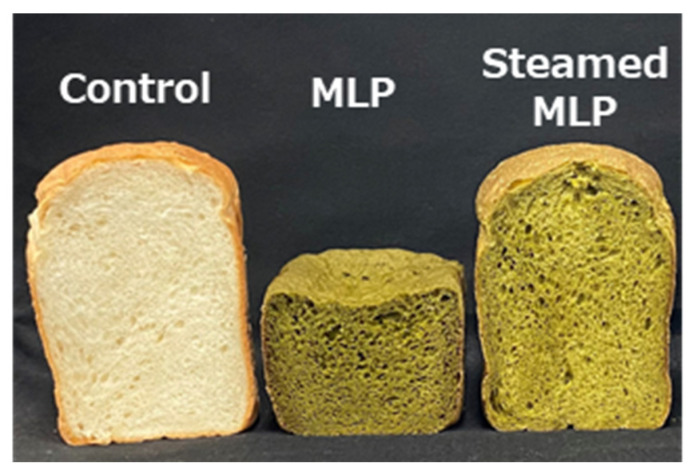
Improvement in bread loaf volume due to the addition of steam-treated moringa leaf powder.

**Figure 3 foods-14-00927-f003:**
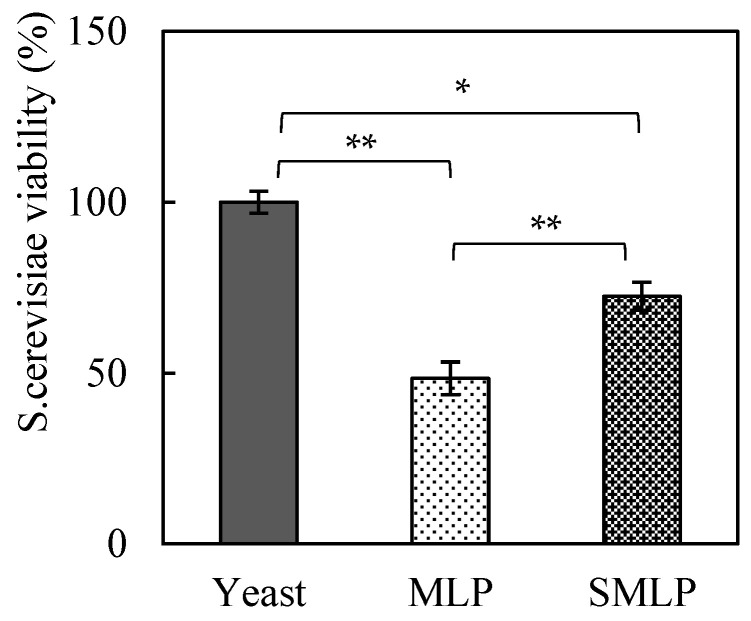
Effect of moringa treatments on *S. cerevisiae* viability. The *S. cerevisiae* survival rate in the control group (no moringa) was 100%. Data analysis was performed using Student’s *t*-test (* *p* < 0.05, ** *p* < 0.01).

**Figure 4 foods-14-00927-f004:**
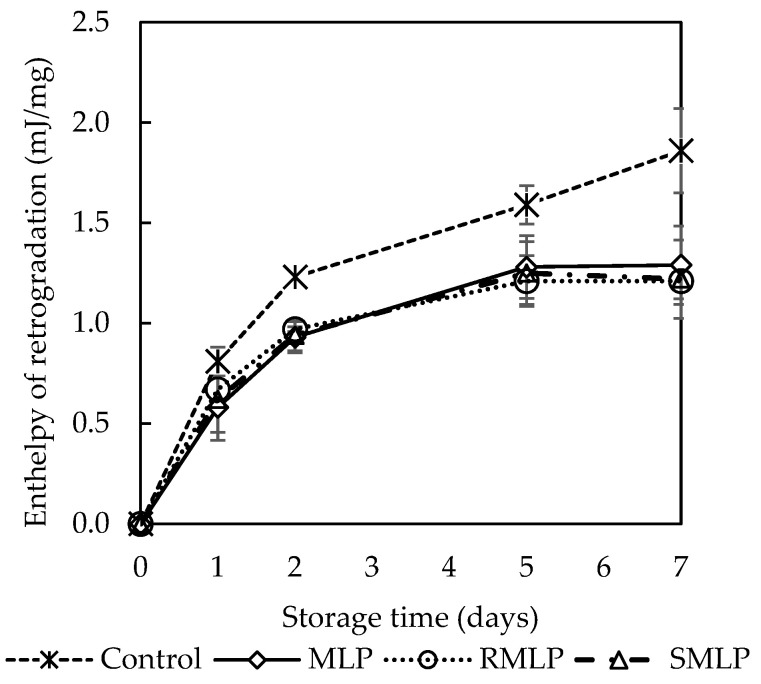
Re-gelatinization enthalpy for rice samples stored at 20 °C for 7 days. The vertical bars represent the standard deviation of each value. Data points followed by different letters indicate significant differences (Tukey’s test, *p* < 0.05).

**Table 1 foods-14-00927-t001:** Chemical composition of wheat flour, *Moringa oleifera* leaf powder (MLP), and steamed MLP (SMLP) on a dry-weight basis (dwb).

Ingredient	Wheat Flour ^(1)^			MLP ^(1)^			SMLP		
Water content (%)	14.3	±	0.4 ^a^	3.7	±	0.5 ^c^	11.7	±	0.3 ^b^
Protein (% dwb)	13.5	±	0.4 ^b^	27.7	±	8.2 ^a^	23.1	±	0.6 ^a^
Carbohydrate (% dwb)	84.6	±	1.1 ^a^	53.2	±	8.0 ^b^	60.2	±	1.0 ^b^
Crude fat (% dwb)	2.6	±	0.8 ^b^	8.8	±	1.1 ^a^	7.8	±	1.3 ^a^
Ash (% dwb)	0.5	±	0.1 ^b^	10.8	±	0.1 ^a^	8.8	±	1.4 ^a^
Energy (kcal/100 g dwb)	420	±	9.8 ^a^	371	±	36 ^b^	404	±	42 ^b^
Na (mg/100 g dwb)	0	±	0.0	127	±	0.7 ^b^	141	±	4.4 ^a^
K (mg/100 g dwb)	102	±	2.4 ^b^	2570	±	26 ^a^	2377	±	15 ^a^
Ca (mg/100 g dwb)	20	±	0.5 ^b^	2024	±	22 ^a^	1780	±	10 ^a^
Mg (mg/100 g dwb)	26	±	0.6 ^b^	470	±	4.3 ^a^	376	±	1.8 ^a^
Fe (mg/100 g dwb)	1	±	0.1 ^b^	10	±	0.5 ^a^	7.4	±	0.2 ^b^

^(1)^ Values for wheat flour and MLP were obtained from our previous study published in Foods (2023) [20]. Values with different letters in the same row are significantly different (Bonferroni test, *p* < 0.05; *n* = 3).

**Table 2 foods-14-00927-t002:** Color and physicochemical properties of wheat flour, moringa leaf powder (MLP), steamed MLP (SMLP), and their respective bread samples.

			Wheat Flour			MLP			SMLP		
**Flour**	Color	L*	78.0	±	0.01 ^a^	55.3	±	0.01 ^b^	41.5	±	0.01 ^c^
		a*	−1.1	±	0.04 ^a^	−7.0	±	0.13 ^c^	−3.6	±	0.04 ^b^
		b*	8.39	±	0.01 ^b^	31.4	±	0.02 ^a^	28.4	±	0.04 ^a^
		ΔE	―			―			15.5	±	0.04
DPPH radical scavenging activity (IC_50_, μg/mL)			N/A			318.9	±	7.11 ^b^	439.9	±	6.71 ^a^
Total phenol content (μg GAE/mL)			N/A			7.0	±	0.24 ^a^	6.0	±	0.16 ^b^
Q3G content (mg/100 mg)			N/A			1.2	±		0.9	±	
**Bread ^(^** ^1)^		Moisture content (%)	45.3	±	0.58	44.7	±	0.01	45.2	±	0.40
		Loaf volume (cm^3^)	1780	±	83.7 ^a^	914	±	6.25 ^b^	1837	±	21.6 ^a^
		Specific volume (cm^3^/g)	4.6	±	0.01 ^a^	2.2	±	0.1 ^b^	4.6	±	0.1 ^a^
		Hardness (N/cm^2^)	0.9	±	0.23 ^c^	4.6	±	0.86 ^a^	0.5	±	0.15 ^b^
		Elasticity	1.2	±	0.25 ^a^	0.1	±	0.01 ^b^	0.9	±	0.12 ^a^
		Chewiness	0.8	±	0.02 ^b^	0.6	±	0.03 ^c^	1.6	±	0.45 ^a^
	VRC ^(2)^%		0.4	±	0.03	0.3	±	0.01	0.3	±	0.02
	Crumb color	L*	78.1	±	0.96 ^a^	41.7	±	2.01 ^c^	51.8	±	1.18 ^b^
		a*	−1.1	±	0.22 ^a^	−1.8	±	0.31 ^b^	−2.1	±	0.67 ^b^
		b*	13.6	±	1.1 ^b^	33.5	±	1.8 ^b^	32.5	±	3.1 ^b^
		ΔE	―			―			10.8	±	1.8
	Crust color	L*	56.4	±	2.27 ^a^	35.7	±	0.33 ^b^	42.8	±	1.7 ^b^
		a*	14.1	±	1.06 ^a^	7.1	±	0.58 ^b^	7.2	±	0.7 ^b^
		b*	32.7	±	1.08 ^a^	21.5	±	0.67 ^b^	24.8	±	3.48 ^b^
		ΔE	―			―			8.2	±	0.90

^(1)^ Bread containing each powder. In all cases, MLP or SMLP substituted 5% of the wheat flour weight. ^(2)^ Volume recovery coefficient. Values with different letters in the same row are significantly different (Bonferroni test, *p* < 0.05; *n* = 5).

**Table 3 foods-14-00927-t003:** Antioxidant activity, total phenolic content, and Q3G concentration of MLP and SMLP.

	DPPH Radical Scavenging Activity (IC_50_, μg/mL)			Total Phenol Content (μg GAE/mL)			Q3G Content (μg/100 mg)		
MLP	318.9	±	7.11 *	7.0	±	0.24 *	89.6	±	0.65 *
SMLP	439.9	±	6.71 *	6.0	±	0.16 *	60.5	±	3.85 *

Data are presented as the mean ± SD (*n* = 3) and were analyzed using Student’s *t*-test (* *p* < 0.01).

**Table 4 foods-14-00927-t004:** Sensory evaluation of bread containing untreated moringa leaf powder (MLP) and steam-treated MLP (MLP and SMLP, respectively).

	Taste			Aroma			Appearance			Texture			Overall		
MLP Bread	5.7	±	1.9	4.9	±	1.6 **	6.2	±	1.5 *	6.6	±	1.7	6.0	±	1.6 *
SMLP Bread	7.0	±	1.4	6.5	±	1.4 **	7.5	±	1.1 *	7.3	±	1.6	7.3	±	1.3 *

Values are reported as the mean ± SD, n = 33. Data analysis was performed using Student’s *t*-test (* *p* < 0.05, ** *p* < 0.01).

## Data Availability

The original contributions presented in this study are included in the article. Further inquiries can be directed to the corresponding author.

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
