# Peer review of "Enhancing Bread Quality with Steam-Treated Moringa (Moringa oleifera) Powder"

_foods, 2025, doi:10.3390/foods14060927_

Round 1

Reviewer 1 Report (Previous Reviewer 1)

Comments and Suggestions for Authors

The authors have expanded the presentation and discussion of the results in accordance with the reviewer's suggestion. I have no further comments on the manuscript.

Author Response

Thank you for your valuable feedback. We appreciate your time and effort in reviewing our manuscript.

Reviewer 2 Report (Previous Reviewer 2)

Comments and Suggestions for Authors

The authors responded to my suggestions and they have addressed all the comments appropriately.

Author Response

Thank you for your valuable feedback. We appreciate your time and effort in reviewing our manuscript.

Reviewer 3 Report (New Reviewer)

Comments and Suggestions for Authors

The manuscript entitled: Enhancing Bread Quality with Steam-Treated Moringa (Moringa oleifera) Powder.

Aimed to investigate the effects of steam-treated Moringa leaf powder (SMLP) on the expansion and sensory qualities of bread supplemented with Moringa leaf powder.

Can steam treatment of Moringa leaf powder improve bread expansion, sensory quality, and functional properties while maintaining its antioxidant benefits? Is the main question.

The topic of the text is relevant and addresses specific gaps in the field, namely, the lack of studies on how processing methods can improve MLP’s functionality in bread-making without compromising its nutritional properties; the limited understanding of the impact of MLP on yeast activity and dough expansion (the study clarified how steam treatment alleviates yeast inhibition); the absence of natural, sustainable methods to counteract MLP’s negative effects on bread texture and sensory properties (the study introduced an eco-friendly solution); no prior research on MLP’s potential to delay bread staling (the study provided new insights into its role in starch retrogradation).

The novelty lies in being the first study investigating steam treatment as a natural processing method to enhance MLP’s baking performance - overcoming its traditional limitations in bread-making.

The study demonstrates improved bread expansion (specific loaf volume increased from 2.2 to 4.6 cm³/g) without chemical additives or fortifiers.

Revealed steam treatment’s ability to mitigate MLP’s inhibitory effects on Saccharomyces cerevisiae viability (improving yeast viability from 48% to 72%), leading to better dough rise.

Preserved MLP’s antioxidant properties while introducing anti-staling benefits (30–40% reduction in starch retrogradation enthalpy), prolonging bread freshness.

Suggested SMLP as a functional ingredient for other bakery products (cakes, cookies, pastries) with potential for industrial-scale food production.

The study fills a critical gap in functional food research by demonstrating that steam-treated MLP (SMLP) can enhance bread-making performance while preserving nutritional benefits, opening doors for its broader application in commercial bakery products.

Overall, the manuscript is structured, clear and easy to read, but I would like to suggest some changes and improvements.

I suggest removing the species name from title and putting it into the abstract in order to have a clean title.

The abstract with less then 200 word is according to the journal's instructions, presents overall main studies and used methods with the principal results and conclusions.

Keywords with 5 of possible 10 words is ok.

The introduction is well-prepared and is relevant to the work.

The methodology is well described, (with only minor incorrections that can be solved).

For each mentioned equipment please add (brand, city, state abbreviation if USA, country) and its operating conditions.

The first time you use an abbreviation please write it out in full firstly.

The equations in the manuscript are numbered according journal rules.

The results and discussion are extensive and well explained, usually, the results obtained are discussed and compared with those from other works

Pictures are ok and Tables need to be formatted according journal rules.

In the tables I suggest to remove the big space between media and standard deviation:

As an example: the authors have 14.3          ±          0.4 and I suggest to put 14.3 ± 0.4 to simplify the tables.

The letter index of significant differences is missing in Table 1 results.

Please change double line top border to single line in table 1, 2 and 3. The same in table 4 for the middle border.

Remove the dot of (n=3.) in table 3 footnote.

In table 4 please consider to put the * and ** near de property and not inside the table with lines. Aroma**   Appearance* Overall*

The conclusions need to be improved showing they are consistent with the evidence and arguments presented and answer the main question posed.

Authors have an Abbreviations section but this section needs to be completed with the missing used manuscript abbreviations.

References are adequate and in 25 papers presented, 1 is from the manuscript's authors.

The references list needs to be reviewed in terms of the formatting.

Species names need to be in italic. The journal title must be in italic but the paper title not. Page numbers.

Please insert DOI in bibliographic references to make it easy to the reader to find them.

Considering the comments, an article revision is recommended.

Author Response

This manuscript is a resubmission of an earlier submission. The following is a list of the peer review reports and author responses from that submission.

Round 1

Reviewer 1 Report

Comments and Suggestions for Authors

The article submitted for review, titled " Enhancing Bread Quality with Steam-Treated Moringa (Moringa oleifera) Powder", requires additional experiments in order to be considered for publication. The experiments presented in the article are interesting, technically well described, statistically processed and correctly discussed. The main problem is the limited number of samples used in the experiment. The research is largely based on and continues the author's previous research, also published in Foods, 2023, 12, 3760, titled “Effects of Roasting on the Quality of Moringa oleifera Leaf Powder and Loaf Volume of Moringa oleifera-Supplemented Bread”. Some parts of the text, especially in the material and methods section, are almost identical to the previous article, as well as part of the results that were taken over, as the authors themselves state. However, the previous research included a much wider change in roasting parameters, which included changing both the time (10, 20 and 30 min) and the heating temperature (110, 130, 150, 170 and 190 degrees Celsius). In this research, the novelty is that instead of roasted Molinga Leaf Powder, Steam-Treated Molinga Powder is used, but there is only one treatment and the only data reported is 10 min in steam mode. I am of the opinion that scientific research should include a much wider interval of changing input parameters in order to draw credible conclusions about the influence and effects in this specific case of steam treatment on the nutritional and qualitative properties of bread supplemented with SMLP.

Considering that in the review form provided by the journal Foods in the Overall recommendation section, under Rejection in parentheses, it is stated that one of the reasons for such a decision is the need for additional experiments, I have made this decision. However, if the authors are willing to supplement the research with additional treatments, results and discussions, I am ready to reconsider my decision and review the text again.

Reviewer 2 Report

Comments and Suggestions for Authors

Dear authors, below are some considerations with suggestions for improving the work

The wording duplication percentage in the manuscript is high and should be reduced.

Limitations of the study: please include the limitations in the results ‘discussion and conclusion if not performed during the study

- The study lacks comparative data with other common natural bread fortifiers

- The paper would benefit from a discussion on the potential antinutritional factors in MLP and SMLP, such as phytates and oxalates, they can affect nutrient bioavailability?

- Please discuss potential limitations, such as the scalability of steam treatment for industrial applications

- Authors considered to perform a shelf-life study? Authors think that this is an important topic?

- An impact of SMLP on the glycemic index of bread could occur?

- The lack of a control for different steaming times is missing. Varying durations into the optimal steaming conditions were considered? And steaming durations and MLP concentrations?

Introduction:

- The hypothesis that steam treatment improves bread quality is logical, but there is a dose-response relationship?

- Please include more recent studies on the use of MLP in food products

- Please add a paragraph discussing the challenges of incorporating MLP into bread such as its impact (if present) on rheology and gluten formation

MM: clarify how the sensory scores were analyzed statistically

Results and discussion: some sections contain long, complex sentences that could be simplified

- When discussing antioxidant activity please clarify the relationship between total polyphenol content and antioxidant activity

- The discussion could be enriched by comparing findings with more recent studies on MLP in baked goods. How do these results compare to other studies on heat-treated plant powders in bread?

- Please elaborate on why SMLP, despite reduced antioxidant activity, still offers functional benefits in bread.

- Why steam treatment did not completely eliminate the inhibitory effects on yeast viability? How this could be addressed in future studies?

- Could be interesting include a discussion of how the findings align with or differ from previous studies on MLP in bread, particularly those that have explored other heat treatment methods.

Conclusion: please include potential applications of SMLP in other bakery products and suggestions for industrial scalability. Results could be useful to consider utilize it in other food products?